# Genome-Wide Identification and Expression Analysis of the PHD Finger Gene Family in Pea (*Pisum sativum*)

**DOI:** 10.3390/plants13111489

**Published:** 2024-05-28

**Authors:** Mingli Liu, Wenju Li, Xiaoling Zheng, Zhuo Yuan, Yueqiong Zhou, Jing Yang, Yawen Mao, Dongfa Wang, Qing Wu, Yexin He, Liangliang He, Dan Zong, Jianghua Chen

**Affiliations:** 1School of Life Sciences, Southwest Forestry University, Kunming 650224, China; liumingli08@163.com (M.L.); wjlicoke@swfu.edu.cn (W.L.); 2CAS Key Laboratory of Tropical Plant Resources and Sustainable Use, CAS Center for Excellence for Molecular Plant Science, Xishuangbanna Tropical Botanical Garden, Chinese Academy of Sciences, Kunming 650223, China; zhengxiaoling@xtbg.ac.cn (X.Z.); yuanzhuo827@gmail.com (Z.Y.); zhouyueqiong@xtbg.ac.cn (Y.Z.); yangjing3007@swfu.edu.cn (J.Y.); maoyawen16@mails.ucas.ac.cn (Y.M.); wangdongfa@xtbg.ac.cn (D.W.); wuqing18@mails.ucas.ac.cn (Q.W.); h199911292022@163.com (Y.H.); 3University of Chinese Academy of Sciences, Beijing 100049, China; 4School of Ecology and Environmental Science, Yunnan University, Kunming 650504, China; 5School of Life Sciences, University of Science and Technology of China, Hefei 230027, China

**Keywords:** *Pisum sativum*, PHD finger, gene family, anther development

## Abstract

The plant homeodomain finger (PHD finger) protein, a type of zinc finger protein extensively distributed in eukaryotes, plays diverse roles in regulating plant growth and development. While PHD finger proteins have been identified in various species, their functions remain largely unexplored in pea (*Pisum sativum*). In this study, we identified 84 members of the PHD finger gene family in pea, which displayed an uneven distribution across seven chromosomes. Through a comprehensive analysis using data from *Arabidopsis thaliana* and *Medicago truncatula*, we categorized the PHD finger proteins into 20 subfamilies via phylogenetic tree analysis. Each subfamily exhibited distinct variations in terms of quantity, genetic structure, conserved domains, and physical and chemical properties. Collinearity analysis revealed conserved evolutionary relationships among the PHD finger genes across the three different species. Furthermore, we identified the conserved and important roles of the subfamily M members in anther development. RT-qPCR and in situ hybridization revealed high expression of the pea subfamily M members *PsPHD11* and *PsPHD16* in microspores and the tapetum layer. In conclusion, this analysis of the PHD finger family in pea provides valuable guidance for future research on the biological roles of PHD finger proteins in pea and other leguminous plants.

## 1. Introduction

Zinc finger proteins are defined by a conserved sequence motif composed of conserved cysteine and histidine residues that bind to zinc atoms in different arrangements [1]. These proteins play diverse biological roles and are present in a range of transcription factors. In plants, the zinc finger domain plays a crucial role in recognizing specific DNA sequences and facilitating protein interactions. Based on the arrangement of amino acids and zinc atoms, zinc finger proteins can be divided into TFIIIA, GATA, LIM, RING, and PHD, as well as newer motifs such as WRKY and Dof. These classifications have evolved through the co-evolution of the original functional motifs and distinct regulatory processes in plants, enabling them to adapt to environmental changes [1,2].

Plant homeodomain finger (PHD finger) proteins contain one or multiple PHD finger domains, which coordinate with two zinc atoms to form a particular structure comprising 50–80 amino acids [3]. According to the specific position of histidine within the cysteine scaffold, the PHD finger domain exhibits a conserved cysteine and histidine arrangement known as Cys4-His-Cys3, which is different from LIM (Cys2-His-Cys5) and RING (Cys3-His-Cys4) [3,4]. The PHD finger was first identified in maize Zmhox1a and *A. thaliana* HAT3.1 proteins by Schindler et al. in 1993 [5,6]. Subsequently, an increasing number of PHD finger proteins have been identified and studied in various species. For instance, there are reportedly 70 PHD finger family members in *A. thaliana* and 33 in *Brassica rapa*, both of which are cruciferous plants [7,8]. Leguminous plants like *Medicago truncatula* were discovered to contain 64 *MtPHDs*, while soybeans (*Glycine max*) contained 95 *GmPHDs* [9,10]. Additionally, the Solanaceae plant *Capsicum annuum* had 57 PHD finger members [11], whereas the Rosaceae plant Chinese pear (*Pyrus bretschneideri*) contained 31 PHD finger members [12]. There were 244 TaPHD genes in wheat (*Triticum aestivum*), 79 in *Sorghum bicolor*, 59 in rice (*Oryza sativa*), and 67 ZmPHD genes in maize (*Zea mays*) among the gramineae [13,14,15,16]. These findings highlight a significant variation in the number of PHD finger family members among monocot and dicot plants, ranging from 31 to 244, indicating a widespread distribution of PHD finger genes that perform diverse functions in plants.

PHD finger proteins play crucial roles in regulating various aspects of plant growth and development, including vernalization, stress response, and anther formation. Vernalization is a key mechanism employed by plants to improve their reproductive success, whereby they perceive the external environment to regulate their flowering time. In *A. thaliana*, the PHD finger gene *VERNALIZATION INSENSITIVE 3* (*VIN3*; a PHD finger gene) initiates *FLC* repression through minimal acetylation, then coordinates with *VERNALIZATION 1* (*VRN1*; a MADS-box gene) and *VERNALIZATION 2* (*VRN2*; encoding a C2H2 transcription factor) to maintain steady *FLC* repression, ultimately facilitating flowering [17,18,19]. During long-term cold exposure, another PHD finger protein VERNALIZATION 5 (VRN5) interacts with VIN3 protein to form a heterodimer complex, resulting in the reduction of *FLC* expression [20]. Furthermore, PHD finger protein plays a key regulatory role in plant responses to stress. Six GmPHD proteins (GmPHD1-6) have been identified in soybean, exhibiting varied responses to NaCl, drought, ABA, and temperature conditions. Overexpression of *GmPHD2* has been found to enhance plant salt tolerance [21]. Among them, *GmPHD5* can interact with methylated H3K4 to regulate the expression of downstream genes associated with salt stress [22]. Several stress-responsive genes have been identified in maize, such as *ZmPHD14*, *ZmPHD19*, and *ZmPHD30*, which could respond to ABA, PEG, and NaCl treatments [14]. Significant alterations in the expression levels of several PHD finger genes in wheat were observed under stress, indicating their potential biological functions in stress responses [16]. In addition, the PHD finger protein family is involved in the regulation of anther development and meiosis in plants. In *A. thaliana*, the *MS1* mutation showed a male sterile phenotype [23]. The mutation of *MMD1* resulted in cytoplasmic shrinkage, collapse of male meiotic cells, and abnormal meiotic products (dyad and triad) [24]. The *duet* is an allelic mutant of *mmd1* and exhibits the same phenotype as *mmd1*, indicating that *MMD1/DUET* is involved in the regulation of chromosome recombination, the meiotic process, and related gene expression during meiosis [25]. *SIZ1*, a SUMO E3 ligase in rice, containing a PHD finger domain, regulates rice fertility by participating in anther dehiscence during anther development [26].

Pea, a leguminous crop, is commonly utilized as the source of grain, vegetable, or forage. Furthermore, it can form a symbiotic relationship with rhizobia, facilitate nitrogen fixation, and reduce the need for fertilizers. This crop plays an important role in food security and sustainable agricultural development, and serves as a valuable model for studying Mendelian genetic theory. The PHD finger family genes regulate plant growth and development. Exploring the PHD finger genes in pea is of great value for agricultural, breeding and the utilization of hybrid vigor. Moreover, the identification and expression analysis of the PHD finger protein family in pea has not been reported. Therefore, this study conducted a genome-wide identification and expression analysis of the PHD finger family in pea, aiming to provide a theoretical reference for studies of the biological function of PHD finger proteins in pea.

## 2. Results

### 2.1. Identification of the PHD Finger Family Genes in Pea

To identify the PHD finger family genes in pea, we initially employed the hidden Markov model of the PHD finger to screen for potential family members. Subsequently, 70 protein sequences of the *A. thaliana* PHD finger gene family were used as query sequences for blast to obtain another set of candidate family members. These two sets of results were combined, and duplicate values were manually removed. CD search and Pfam were combined to further confirm the completeness of domains of the PHD finger gene family members in pea, and the sequences with missing or incomplete PHD finger domains were removed. Ultimately, we identified 84 PHD finger family members in pea, designated as PsPHD1-PsPHD84 based on their chromosomal position (Appendix A).

To further explore the potential biological roles of the PHD finger family genes in pea, we conducted predictions on the physicochemical properties of the 84 PsPHDs. The results revealed important insights into these proteins. The length of amino acids in the PHD finger family ranged from 192 to 2321 aa, while the molecular weight varied from 21.59 to 258.54 KDa. Additionally, the isoelectric point spanned from 4.45 to 9.58. Notably, all identified PsPHDs displayed hydrophilic characteristics. Subcellular localization analysis of PsPHD proteins revealed that PsPHD1 and PsPHD32 were predicted to be located in both the cytoplasm and nucleus, while all the other proteins were predicted to be located solely in the nucleus. Compared with certain functionally conserved gene families, the physicochemical properties of PsPHDs exhibit significant differences. Given the significance of the PHD finger family, we also scrutinized the distribution data of the PHD finger family in other plant species (Table 1). The results revealed significant differences in the distribution and protein properties of the PHD family across different species, providing compelling evidence for the functional diversity of the PHD finger family.

### 2.2. Amino Acid Multi-Sequence Alignment of the PHD Finger Proteins in Pea

In order to further understand the sequence similarity and motif conservation among the PHD finger proteins in pea, we conducted multiple sequence alignment (Figure 1). The results of the multiple sequence alignment revealed that the regions highlighted by red shading represent Cys residues and those highlighted by blue shading represent His residues (Figure 1A). Additionally, we generated a domain sequence logo for 84 genes in the pea PHD finger family (Figure 1B). The results show that the pea PHD finger domain contains a conserved Cys4-His-Cys3 motif (CCCCHCCC), which is responsible for binding to Zn^2+^ and maintaining the stability of the PHD finger domain. Furthermore, the number of bases between CCCCHCCC residues displayed a relatively conserved pattern, consistent with earlier findings [12].

### 2.3. Chromosomal Localization of PsPHDs

The 84 PHD finger genes were mapped to the chromosomes of pea, and the results showed that 83 PHD finger genes were unevenly distributed across the seven chromosomes of pea, while only *PsPHD84* was located on the scaffold (Figure 2). Each chromosome contained a different number of PHD finger genes, ranging from 8 to 18. Notably, chromosome 5 exhibited the highest distribution with 18 genes, whereas chromosome 4 had the lowest distribution with only 8 genes. Furthermore, the PHD finger genes on chromosomes 2 and 3 are predominantly located at the upper ends, whereas chromosomes 4 and 6 displayed a distribution pattern at both ends. Interestingly, on chromosome 7, the majority of the PHD finger genes were found near the base of the chromosome.

### 2.4. Phylogenetic Analysis of the PHD Finger Family in Pea

The classification of protein subfamilies allows for the analysis of the function and evolutionary relationship of PHD finger proteins in pea, as well as their comparison with PHD finger proteins in other species. Seventy full-length sequences of *A. thaliana* and sixty-four full-length sequences of *M. truncatula* PHD finger family proteins were extracted from TBtools. Subsequently, 84 full-length sequences of the pea PHD finger proteins were used to build a phylogenetic tree (Figure 3). The results showed that, in accordance with the subfamilies division guidelines of *A. thaliana*, the PHD finger proteins from pea, *A. thaliana*, and *M. truncatula* were divided into 20 subfamilies (Appendix A). The varying member counts within each subfamily indicate significant functional diversification of the PHD finger family during evolution, showing their involvement in the regulation of distinct biological processes. 

Within the phylogenetic tree, subfamily O has the largest number of members, comprising 27 members, while subfamily C is the smallest, with only 4 members. Interestingly, only *A. thaliana* proteins were clustered in subfamily A; no genes of pea or *M. truncatula* were clustered in it.

### 2.5. Conserved Domain and Gene Structure Analysis of the PHD Finger Family in Pea 

To analyze the potential biological functions of the PHD finger family in pea based on protein structure properties, we utilized a CD search tool to analyze the domain composition of PHD finger proteins. The results were visualized and summarized using TBtools (Figure 4).

In the conserved domain analysis, it was observed that the PHD finger proteins in pea contained 1–4 PHD finger domains. Most proteins predominantly contained only 1 PHD finger domain, while a few proteins contained 2–3 PHD finger domains. Notably, the PsPHD43 protein was the only one found to have four PHD finger domains (Figure 4A). Proteins clustered within the same branch also exhibited significant similarities in the quantity and arrangement of the PHD finger domains. For instance, the seven genes, *PsPHD7*, *PsPHD26*, *PsPHD28*, *PsPHD34*, *PsPHD51*, *PsPHD52*, and *PsPHD8*2, clustered together with the AL subfamily of *A. thaliana* in the L subfamily. This subgroup is characterized by an Alfin domain at the N-terminus and a PHD finger domain at the C-terminus, similar to AL protein domains in other species (Figure 4A). This implies that AL-related genes in pea may also be involved in the response to abiotic stress in plants [2].

In addition to the PHD finger domain, a CD search was also used to identify other domains in the PHD finger proteins of pea; for example, the SET domain, initially discovered in plants with 37 members identified in *A. thaliana*. It functions as a histone lysine methyltransferase and is crucial for maintaining enzyme activity, participating in histone tail modification to regulate gene expression and chromatin structure [27,28]. The PWWP domain has the ability to bind to DNA and methylated histones, playing a crucial role in cell division, growth, and differentiation. Additionally, PWWP domains usually coexist with PHD finger-like zinc finger domains to function, such as *PsPHD53*, *PsPHD33*, *PsPHD19*, *PsPHD20*, *PsPHD27*, and *PsPHD41* [29]. The Jas domain plays a role in plant growth, development, and response to stress environments [30,31], while the Alfin domain exists in the Alfin-like protein family (AL1-AL7), which plays a role in plant stress resistance [32,33,34]. Other domains, such as RING, AAA, BAH, etc., were also identified. Based on the findings from the CD search and Pfam identification, multiple types of domains within the PHD finger proteins of pea carry out single or coordinated roles. These domains are involved in various processes, including plant growth and development, chromatin regulation, and gene transcription regulation. Therefore, the diverse functions of the PHD finger proteins offer valuable insights for further exploration into their roles.

In the gene structure analysis of the PHD finger family in pea (Figure 4B), the number of exons in these genes of pea varies from 1 (*PsPHD68*) to 33 (*PsPHD1, PsPHD32,* and *PsPHD17*). The PHD finger family genes exhibit considerable variation in terms of exon and intron number and arrangement; however, those belonging to the same subfamily exhibit similar patterns. For example, members of the L subfamily (*PsPHD28*, *PsPHD82*, *PsPHD26*, *PsPHD52*, *PsPHD7*, *PsPHD34*, and *PsPHD51*) and members of the Q subfamily (*PsPHD36*, *PsPHD49*, *PsPHD31*, and *PsPHD46*) share a comparable intronexon distribution pattern. 

### 2.6. Collinearity Analysis of PsPHDs

To predict the potential functions of the PHD finger family genes in pea, we constructed collinearity analysis among four species: two dicotyledonous plants (*A. thaliana* and *M. truncatula*) and two monocots (rice and maize).

The results of collinearity analysis indicated the presence of pairwise homologues of the PsPHD genes, with 48, 93, 9, and 6 pairs of homologous genes detected in *A. thaliana*, *M. truncatula*, rice, and maize, respectively (Figure 5). This implies that the evolutionary relationship and homology of PHD finger genes in pea differ significantly between monocot and dicot plants, with PsPHD genes showing a strong evolutionary relationship with *A. thaliana* and *M. truncatula.* The PHD finger genes on each chromosome of pea can produce collinearity with the genomes of *A. thaliana* and *M. truncatula*, with stronger evolutionary relationships observed between pea and *M. truncatula* compared to *A. thaliana* and pea (Figure 5A). Among the monocot plants, the PHD finger genes located on chromosomes 1 and 4 of pea did not exhibit collinearity with the genomes of rice and maize (Figure 5B), indicating that the PHD finger genes in pea, *A. thaliana*, and *M. truncatula* are more evolutionarily conserved during the evolution of higher plants.

### 2.7. Cis-Acting Element Analysis of PHD Finger Family Gene Promoters in Pea

Cis-acting elements, serving as the binding sites for transcription factors, can elucidate their functions through the prediction of cis-acting elements in the promoter region. This analysis establishes a theoretical foundation and a starting point for understanding the functions of the *PsPHDs*. We extracted the 2 kb sequence upstream of the *PsPHD* initiator codon for the analysis of cis-acting elements in the promoter region (Figure 6). The results showed the presence of 25 cis-acting elements, which can be divided into plant hormone response, stress response, and plant growth and development (Appendix A). Notably, the functional categories related to plant growth and development encompass key elements such as light response, regulation of zein metabolism, and meristem expression. Within this category, the most abundant cis-acting elements are associated with light responses, including Box 4, G-Box, GT1-motif, GATA-motif, and TCT-motif (Figure 6). Furthermore, the analysis identified elements related to plant hormone response, including the MeJA responsiveness, salicylic acid response, gibberellin responses, and auxin response elements. For instance, the ABRE element participates in abscisic acid regulation, while elements such as the TGA, AuxRR-core, TGA-box, and AuxRE are associated with auxin regulation. Additionally, GA response elements mainly include TATC-box, Gra-motif, and P-box, while elements like CGTCA-motif and TGACG-motif are involved in the MeJA responsiveness regulation pathway. The regulation of the salicylic acid response mainly depends on the TCA element and the SARE element (Figure 6). Considering environmental stress, certain genes within the PHD finger family in pea respond to stressors such as drought, low temperature, anaerobic induction, and wounds through elements like MBS, LTR, ARE, WUN-motif, and others. Furthermore, elements such as MSA-like, HD-Zip 1, GCN4-motif, AACA-motif, CAT-box, NON-box, and others play important roles in plant growth and development. 

Pollen development is a crucial process in seed plants for successful sexual reproduction and alternation of generation, which is related to crop production and the utilization of heterosis. In *A. thaliana*, the PHD finger domain of *MMD1* recognizes H3K4me2/3 and regulates the expression of genes involved in meiosis. *MS1*, encoding a PHD finger protein, plays an important role in the formation of pollen exine, thereby affecting pollen fertility. In the phylogenetic tree, both *MMD1* and *MS1* were clustered into the M subfamily. To analyze the potential biological functions of the M subfamily genes, we calculated the number of promoter cis-acting elements. Consistent with previous results, the analysis revealed that the highest number of cis-acting elements were associated with light responsiveness, followed by MeJA responsiveness and anaerobic induction response (Figure 7), indicating that the PHD finger family genes in pea may be involved in multiple signal transduction pathways. Based on the analysis of promoter cis-acting elements, it can be inferred that *PsPHDs* play an important role in plant growth and development, hormone regulation, light response, and stress (Appendix A). 

### 2.8. Expression Patterns of the PsPHDs during Anther Development

Understanding the spatial expression patterns of genes provides valuable insights into their functions. In order to further explore the role of the *PsPHDs* in anthers development, we conducted an expression analysis using several *PsPHD* expression datasets from different tissues. It was found that the *PsPHDs* were expressed in most tissues, but the level of expression was different (Figure 8B). To validate the expression levels of specific PsPHD genes in different tissues, we performed RT-qPCR experiments using gene-specific primers. The results showed that the expression pattern of these genes in various tissues of the pea was basically consistent with the above results (Figure 8C–G). *PsPHD6* showed ubiquitous expression across all tissues, while *PsPHD35* and *PsPHD78* displayed specific expression in roots. *PsPHD11* showed specific expression in the reproductive shoot apical meristem and roots, whereas *PsPHD16* was specifically expressed in flower buds during the reproductive stage. Notably, *PsPHD11* and *PsPHD16* were homologous to the *A. thaliana* anther development-related genes *MS1* and *MMD1/DUET* and clustered together in the evolutionary tree., These findings revealed that *PsPHD11* and *PsPHD16* may play a crucial role in the regulation of anther development in pea.

We conducted in situ hybridization to further analyze the expression patterns of *PsPHD11* and *PsPHD16* during anther development in pea. Our results revealed that *PsPHD11* and *PsPHD16* were specifically expressed in microspores and the tapetum in pea (Figure 9), which was consistent with the results of RT-qPCR analysis. Taken together, these findings collectively support the notion that *PsPHD11* and *PsPHD16* may regulate anther development in pea.

## 3. Discussion 

The PHD finger proteins are a class of zinc finger proteins that are widely present in eukaryotes. The PHD finger domain primarily regulates gene expression by recognizing various types of histone modifications and binding to DNA sequences, which plays an important role in vernalization, pollen development, and response to stress [35]. The PHD finger domain is relatively conserved in different plants and contains a conserved cysteine-rich motif (Cys4-His-Cys3). The number of amino acids between cysteine residues and between cysteine and histidine residues is conserved, and the amino acids in front of the final cysteine are also conserved, typically tryptophan or another aromatic amino acid, such as tyrosine or phenylalanine [3].

The function of the PHD finger proteins is also conserved in different species. *MS1* is a key gene that regulates tapetum and microspore development in *A. thaliana*. Mutations in *MS1* result in complete male sterility. Similarly, mutations in its homologues *PTC1*, *ZmMS7*, and *MS9* in rice, maize, and sorghum, respectively, also lead to the same phenotype [23,36,37,38]. Another important gene, *MMD1/DUE*T, is involved in chromosome recombination and the meiotic process during meiosis in *A. thaliana* [25]. Previous studies have shown that the PHD finger domain can bind to modified or unmodified C-terminal tails of histones H3 and H4 [39]. In the case of *MMD1/DUET,* it recognizes and binds to H3K4me2, participating in chromatid separation and cytokinesis during the meiosis II. Additionally, MMD1/DUET regulates the expression of the *TDM1* by binding to its promoter region [40]. Moreover, *MMD1/DUET* activates the expression of the *CAP-D3* to facilitate the chromosome reaching a highly coendensed state [41]. Further studies on the molecular mechanism of condensation have revealed that MMD1/DUET interacts with histone H3K4 demethylase JMJ16, enabling it to demethylate H3K9me3 and promote the progress of meiotic chromosome condensation [42]. Interestingly, *MS4*, which is a direct homologue of the MMD1/DUET in dicot soybean, and *TIP3*, a closely related gene in monocot rice, also play crucial roles in the regulation of anther and microspore development. These genes displayed a similar male sterile phenotype after mutation, indicating that the PHD finger proteins possess conserved functions in regulating the development of anthers and pollen in both dicotyledonous and monocotyledonous plants [25,43,44]. In our study, *PsPHD11* and *PsPHD16* in pea belonged to subfamily M and clustered together with the *MMD1/DUET* and *MS*1. Subsequently, we performed RT-qPCR analysis of the PsPHD genes in the subfamily M and observed that *PsPHD11* and *PsPHD16* genes exhibited high expression levels at the shoot apical meristem during the reproductive stage. Further in situ hybridization analysis found that both *PsPHD11* and *PsPHD16* genes are expressed in the tapetum and microspores of pea anthers, indicating that *PsPHD11* and *PsPHD16* genes potentially participate in the regulation of the meiosis process and pollen development during pea reproduction. The PHD finger proteins have a certain conservation through the process of evolution, which requires validation by further studies. Certain subfamilies’ genes exhibit conserved domains and gene structures, which reveals that their biological roles may be comparable. Previous studies have indicated that the ING subfamily (*AtING1*, *AtING2*) and AL subfamily (AL1-7) in *A. thaliana* belong to the PHD finger family. Members of these subfamilies contain only one PHD finger domain [45]. Within the AL subfamily of *A. thaliana*, *AL6* is involved in root hair elongation under phosphorus starvation, while *AL5* can respond to abiotic stresses (such as high salt, drought, and cold stress), and overexpression can improve its stress resistance [2,46]. In the phylogenetic tree, the *PsPHD7*, *PsPHD26*, *PsPHD28*, *PsPHD34*, *PsPHD51*, *PsPHD52*, and *PsPHD82* are clustered together with seven members of the AL subfamily (AL1-7) of *A. thaliana*, forming the L subfamily. Showing these genes may also play vital roles in the response to abiotic stress in pea. *M. truncatula* and pea are commonly used as model plants of legumes to study flowering time. *MtING2* regulates the flowering time and growth of *M. truncatula* [47]. Phylogenetic analysis indicates a close evolutionary relationship between *MtING2 (MTR-7g085450)* and *PsPHD43* in pea, implying that *PsPHD43* may play a potential role in the regulation of flowering time in pea, warranting further study and exploration in subsequent research steps.

We conducted a comprehensive biogenic analysis of 84 genes within the PHD finger family in pea, which were categorized into 20 subfamilies based on *A. thaliana* subfamily classification criteria. In the multi-species collinearity analysis, pea exhibited a stronger evolutionary relationship with the dicots *A. thaliana* and *M. truncatula* compared to the monocots rice and maize. Given that both pea and *M. truncatula* are leguminous plants, the PHD finger genes of pea and *M. truncatula* are more evolutionarily conserved. Analysis of cis-acting elements in the promoter region indicated that *PsPHDs* may participate in the regulation of biological processes such as growth and development, hormone regulation, light response, and stress response.

In the phylogenetic analysis, the *PsPHD11* and *PsPHD16* genes clustered together with *MMD1/DUET* and *MS*1, which are involved in the development of anthers and pollen in *A. thaliana* [23,24,25]. RT-qPCR and in situ hybridization analysis showed that both *PsPHD11* and *PsPHD16* exhibit expression in the tapetum and microspores of the anther in pea, indicating the *PsPHD11* and *PsPHD16* may be involved in the regulation of the meiosis process and pollen development. The conservation of the PHD finger proteins during evolution requires further verification through additional studies. In conclusion, the above results serve as a reference for further studies into the functions of the PHD finger family genes in pea.

## 4. Materials and Methods

### 4.1. Identification of the PHD Finger Family Members in P. sativum

We downloaded the pea genome data, proteome data, and GFF files from the Pea Genome Database (https://www.peagdb.com). Initially, we obtained the hidden Markov model (.HMM file) and Pfam number of the PHD finger from the Pfam website (http://pfam.xfam.org/), and preliminarily screened using local hmmsearch software (v3.0) and the simple HMM search in TBtools (v2.069), respectively. The results were merged, and duplicate values were removed to identify candidate family members. Subsequently, we used the protein sequences of 70 members of the PHD finger gene family of *A. thaliana* as query sequences and obtained candidate family members through blast using TBtools (v2.069). The identified members from the two methods were combined, including some redundant and repetitive members. Consequently, the repetitive sequences were manually deleted to obtain the final candidate members of the pea PHD finger gene family. In order to further elucidate the family members, we utilized the CD Search Tool on NCBI and Pfam analysis to eliminate the sequences with missing or incomplete PHD finger domains. As a result, we determined that the pea PHD finger family comprises 84 members. We employed the online tool Cell-PLoc (http://www.csbio.sjtu.edu.cn/bioinf/Cell-PLoc-2/) to predict the subcellular localization. TBTools (v2.069) was also used to predict protein physical and chemical properties, including amino acid number, molecular weight, isoelectric point, instability index, and hydrophilicity.

### 4.2. Chromosome Location, Amino Acid Sequence Alignment, and Phylogenetic Analysis of the PHD Finger Gene Family Members in Pea

The protein sequences of the PHD finger domain from the 84 members of the pea PHD finger family were extracted due to significant differences in their protein sequences. Subsequently, sequence alignment was performed using MEGA 11 and then visualized using Jalview. TBtools (v2.069) was used to determine the chromosome locations of the pea PHD finger family members, and a phylogenetic tree was constructed using PHD finger proteins from *A. thaliana* (70 proteins), *M. truncatula* (64 proteins), and pea (84 proteins). The bootstrap value was set to 1000, and the phylogenetic tree was categorized into subfamilies based on the subfamilies division standard of *A. thaliana*. In addition, the evolutionary tree of the PHD finger proteins was completed through IQ-Tree Wrapper of TBtools (v2.069). The evolutionary tree was then visually enhanced using Chiplot (TVBOT (chiplot.online) [48].

### 4.3. Conserved Domains and Gene Structure Analysis of the PHD Finger Gene Family Members in Pea

The CD Search Tool on NCBI (https://www.ncbi.nlm.nih.gov/Structure/cdd/wrpsb.cgi) was utilized to predict conserved domains, with the E-Value set to 10e^−5^. TBtools (v2.069) was then employed to visualize the conserved domains and gene structure of the PHD finger gene family members in pea.

### 4.4. Collinearity Analysis of the PHD Finger Gene Family Members among Pea and Other Species

To conduct collinearity analysis among pea and other species, GFF files for *M. truncatula*, *A. thaliana*, rice, and maize were downloaded from the Ensemble Plants website (https://plants.ensembl.org/). Subsequently, TBtools (v2.069) was used to analyze and visualize the collinearity relationships between pea, *A. thaliana*, *M. truncatula*, rice, and maize.

### 4.5. Cis-Acting Elements Analysis of the PHD Finger Members in Pea

For cis-acting elements analysis of the PHD finger members in pea, the gene promoter region (2000 bp before the start codon) was extracted using TBtools (v2.069) and submitted to the online website PlantCARE to predict cis-acting elements (https://www.plantcare.co.uk/). Additionally, TBtools (v2.069) was used to predict and visualize the potential biological functions of the PHD finger family members in pea.

### 4.6. Analysis of the Expression Pattern of the PHD Finger Genes in Different Tissues of Pea

To understand the gene function of the PHD finger genes in pea another development, the electronic expression profile of the PHD finger gene was analyzed [35], and the resulting heat map was visualized using TBtools (v2.069). Furthermore, total RNA was extracted from various tissues of pea for the RT-qPCR experiment using a RnaEx™ Total RNA isolation kit (GENEray Biotech, Shanghai, China), including the root, stem, leaf, vegetative shoot, reproductive shoot, <0.5 cm flower bud, 0.5–1 cm flower bud, 1–1.5 cm flower bud, 1.5–2 cm flower bud, 2–2.5 cm flower bud, fruit pod, and seed. The first-strand cDNA was reverse-transcribed with the HiScript®II 1st Strand cDNA Synthesis Kit (+gDNA wiper) (Vazyme, Nanjing, China). Quantitative real-time PCR was conducted using Magic SYBR Green qPCR Mix (Magic-bio, Hangzhou, China) on a LightCylcer 480 device (Roche, Basel, Switzerland). *PsEF1a* was used as the internal reference gene for RT-qPCR, and the primers are given in Appendix A. In situ hybridization was employed for further analysis of the expression pattern of the two genes highly expressed in the shoot apical meristem during the reproductive stage. *PsPHD11* and *PsPHD16* probes against full-length complementary DNAs were used. Eight-micrometer sections from shoot apices of six-week-old plant materials were conducted and hybridized with digoxigenin-labelled antisense probes. The signals were visualized with an Olympus BX63 microscope.

## Figures and Tables

**Figure 1 plants-13-01489-f001:**
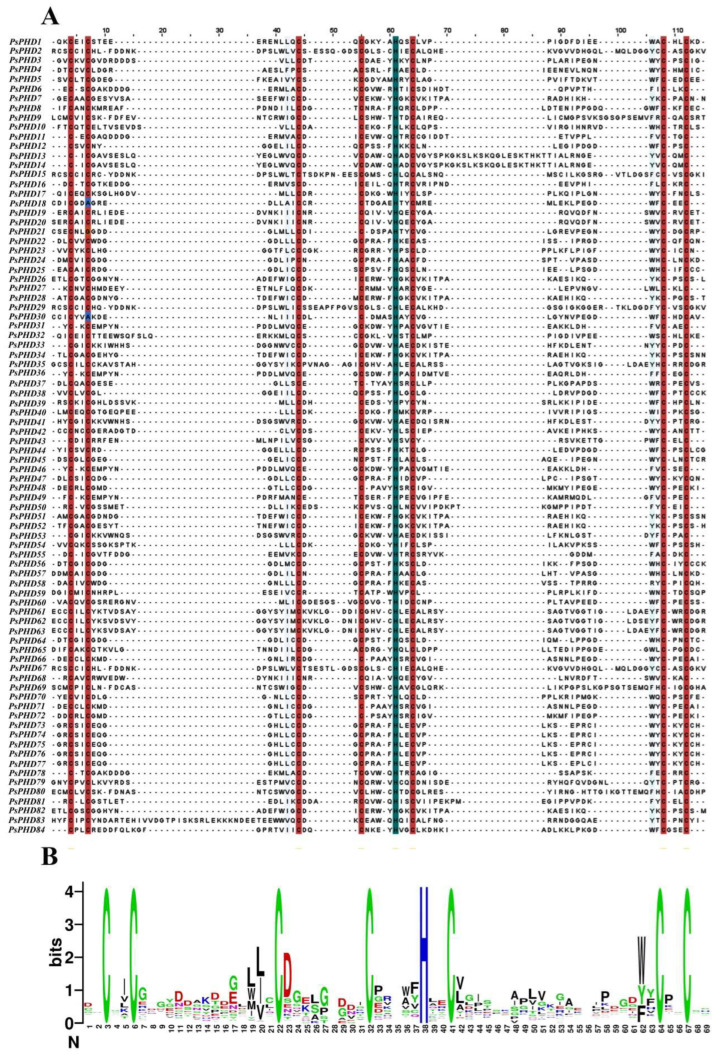
Multiple sequence alignment of domain proteins of the pea PHD finger family members. (**A**) In the figure, the areas with red shading correspond to Cys residues, while the areas with blue shading correspond to His residues. (**B**) Sequence logo of the PHD finger domains in pea; the height of the letters in the figure is proportional to the frequency of occurrence of amino acid residues; bits are often used as units.

**Figure 2 plants-13-01489-f002:**
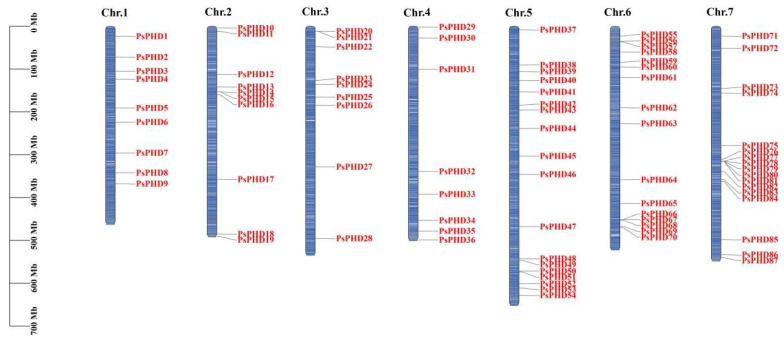
Chromosomal localization of PHD finger genes in pea. On the left is the scale bar, which represents the chromosome length in Mb. Blue bars represent the chromosomes, and the chromosome numbers are shown at the top of the bars.

**Figure 3 plants-13-01489-f003:**
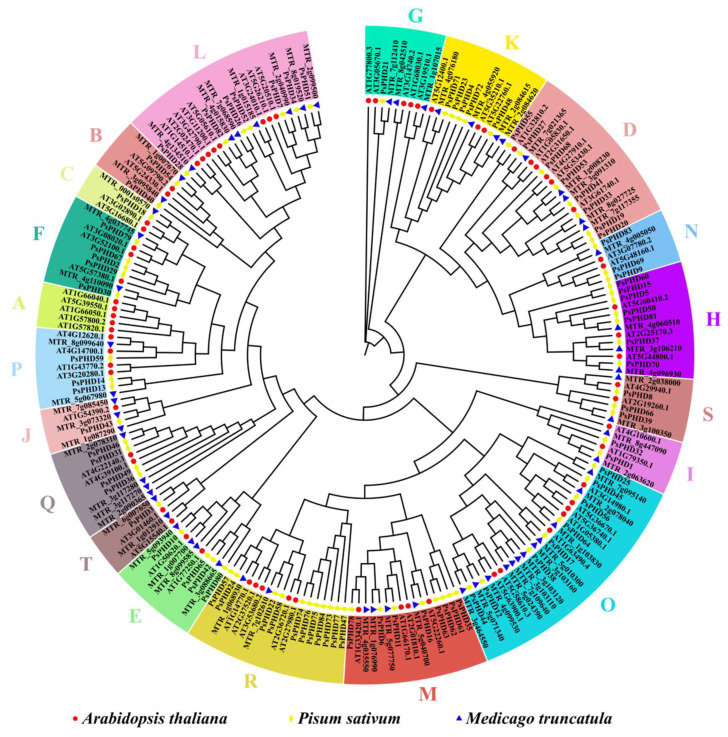
Phylogenetic analysis of the PHD finger proteins in pea and other species. The phylogenetic analysis of the PHD finger family in *P. sativum*, *A. thaliana*, and *M. truncatula* was constructed using TBtools. Based on the standard of *A. thaliana* subfamily division, which categorized them into 20 subfamilies (A–T) distinguished by distinctive colors, the PHD finger proteins from *P. sativum*, *A. thaliana*, and *M. truncatula* are distinguished and represented by yellow diamonds, red circles, and blue triangles, respectively.

**Figure 4 plants-13-01489-f004:**
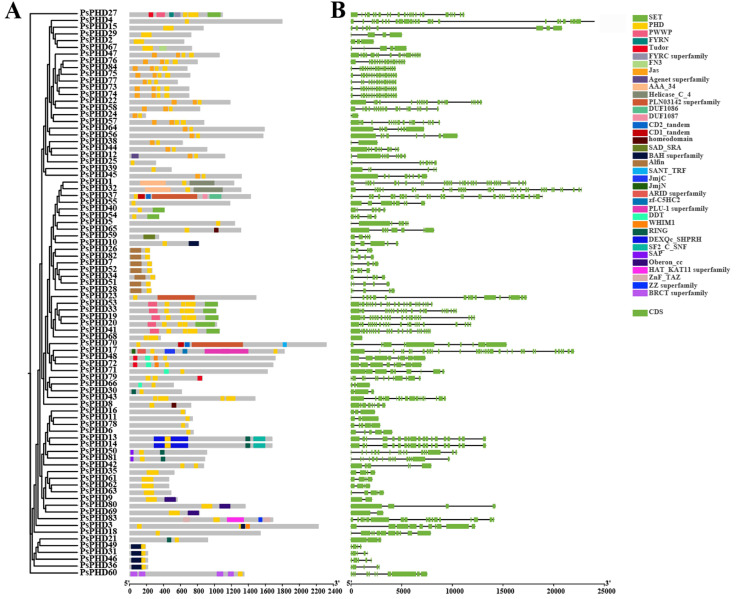
Domain and gene structure analysis of the PHD finger family in pea. (**A**) Evolutionary tree of the PHD finger family proteins and analysis of conserved domains of *PsPHDs*; boxes with different colors represent distinct domains. (**B**) Structural analysis of the PHD finger genes in pea, where green boxes represent CDS.

**Figure 5 plants-13-01489-f005:**
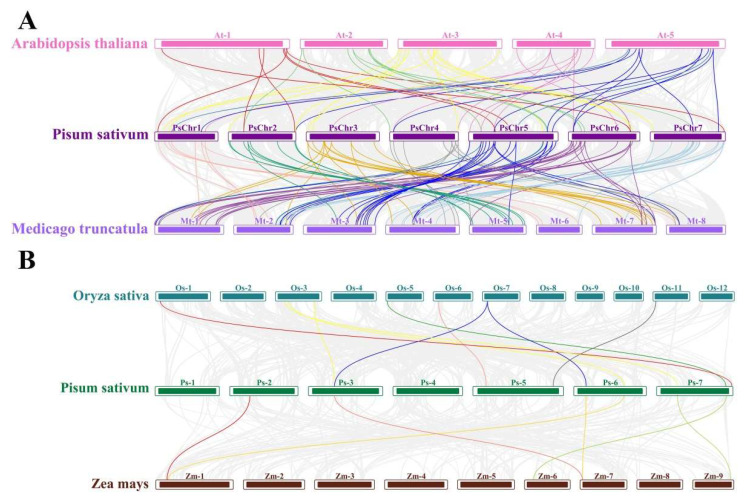
Collinearity analysis of the PHD finger family genes in pea. (**A**) Collinearity analysis of pea with *A. thaliana* and *M. truncatula*. (**B**) Collinearity analysis of pea with rice and maize. Gray lines in the background indicate collinear blocks in the pea and other species’ genomes. Boxes of different colors represent chromosomes of the species, while lines of different colors show gene pairs homologous to the pea PHD finger genes.

**Figure 6 plants-13-01489-f006:**
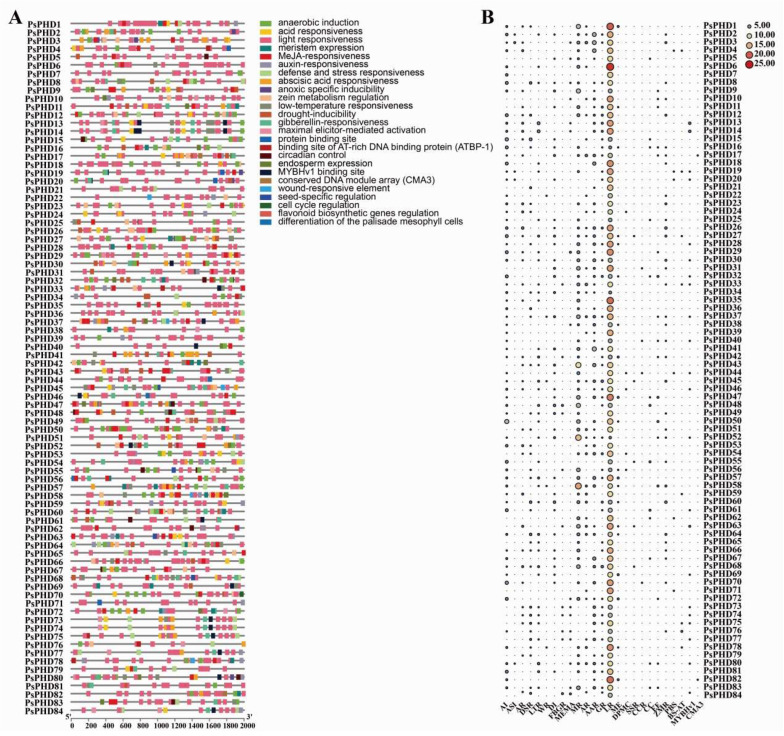
Cis-acting element analysis of *PsPHDs.* (**A**) The 2 kb upstream region of the start codon was analyzed with PlantCARE to identify promoter cis-acting elements, and the results were visualized using TBtools. Diverse promoter cis-acting elements are present in different colored boxes; (**B**) Heat map of promoter cis-acting element analysis of the PHD finger genes in pea. AI: anaerobic induction; ASI: anoxic-specific inducibility; AR: acid responsiveness; DSR: defense and stress responsiveness; LTR: low-temperature responsiveness; WR: wound-responsive element; DI: drought inducibility; FBGR: flavonoid biosynthetic gene regulation; MEMA: maximal elicitor-mediated activation; MR: MeJA responsiveness; AR: auxin responsiveness; AAR: abscisic acid responsiveness; GR: gibberellin responsiveness; LR: light responsiveness; ME: meristem expression; DPMC: differentiation of the palisade mesophyll cells; SSR: seed-specific regulation; CCR: cell cycle regulation; CC: circadian control; EE: endosperm expression; ZMR: zein metabolism regulation; PBS: protein binding site; BS-AT: binding site of AT-rich DNA binding protein (ATBP-1); MYBHv1: MYBHv1 binding site; CMA3: conserved DNA module array (CMA3).

**Figure 7 plants-13-01489-f007:**
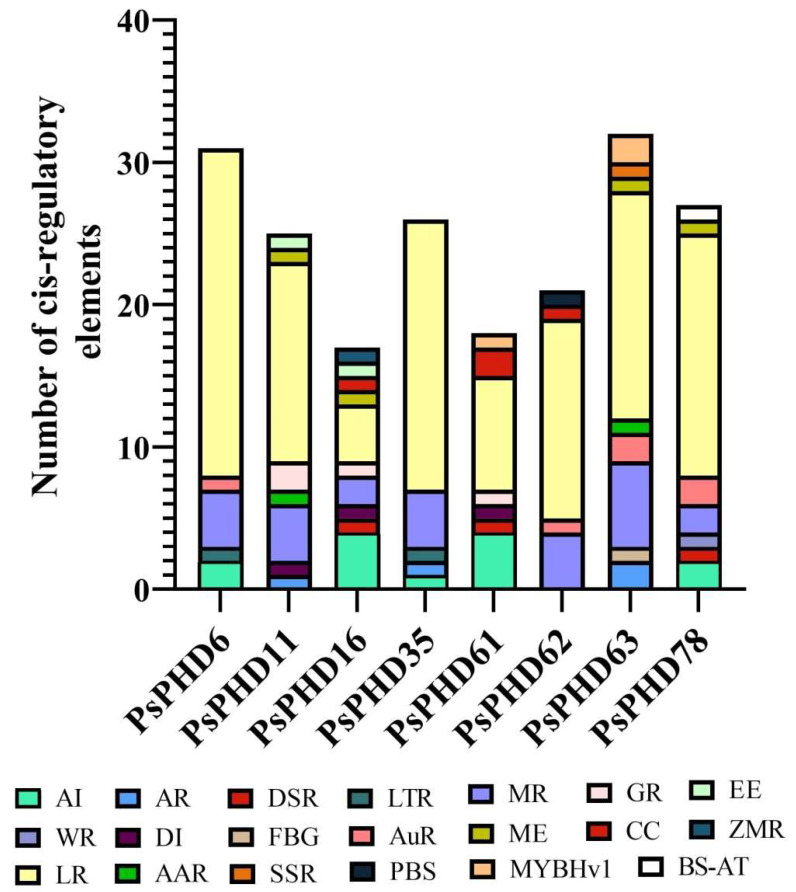
Statistics of cis-acting elements in several genes of the PHD finger family in pea. AI: anaerobic induction; AR: acid responsiveness; DSR: defense and stress responsiveness; LTR: low-temperature responsiveness; WR: wound-responsive element; DI: drought inducibility; ME: meristem expression; AuR: auxin responsiveness; MR: MeJA responsiveness; AAR: abscisic acid responsiveness; FBG: flavonoid biosynthetic gene regulation; LR: light responsiveness; GR: gibberellin responsiveness; EE: endosperm expression; CC: circadian control; BS-AT: binding site of AT-rich DNA binding protein (ATBP-1); ZMR: zein metabolism regulation; SSR: seed-specific regulation; MYBHv1: MYBHv1 binding site.

**Figure 8 plants-13-01489-f008:**
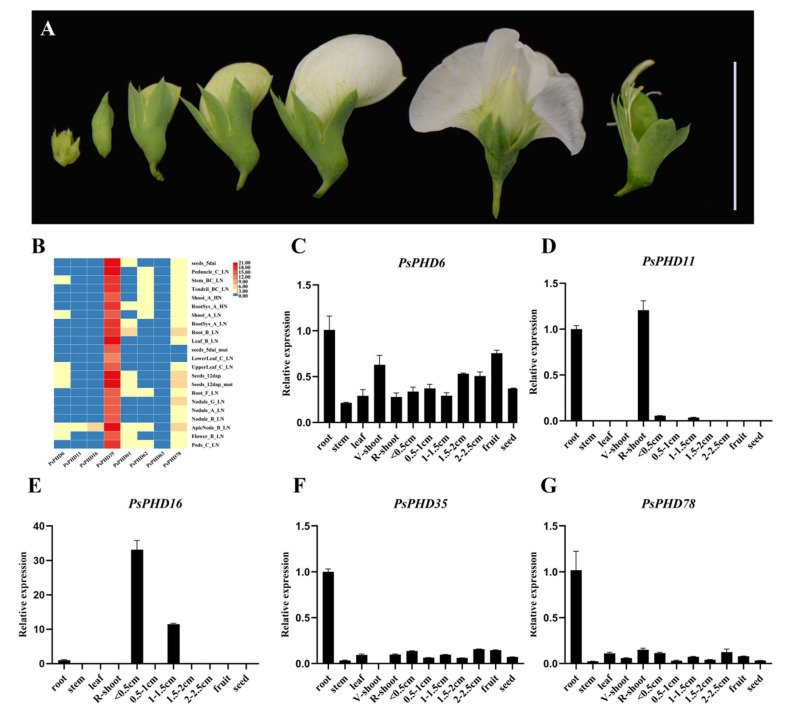
RT-qPCR analysis of the PHD finger genes of pea in different tissues. (**A**) Schematic diagram of several tissues of pea, including reproductive shoot apical meristem, flower buds < 0.5 cm, 0.5–1 cm, 1–1.5 cm, 1.5–2 cm, 2–2.5 cm in length, and capsule. Scale bar = 2 cm. (**B**) Heat map of the *PsPHD6*, *PsPHD11*, *PsPHD16*, *PsPHD35*, *PsPHD61*, *PsPHD62*, *PsPHD63*, and *PsPHD78* gene transcript levels. (**C**–**G**) RT-qPCR analysis of the expression levels of the *PsPHD6*, *PsPHD11*, *PsPHD16*, *PsPHD35*, and *PsPHD78* in different tissues. V-shoot: vegetative shoot apical meristem; R-shoot: reproductive shoot apical meristem.

**Figure 9 plants-13-01489-f009:**
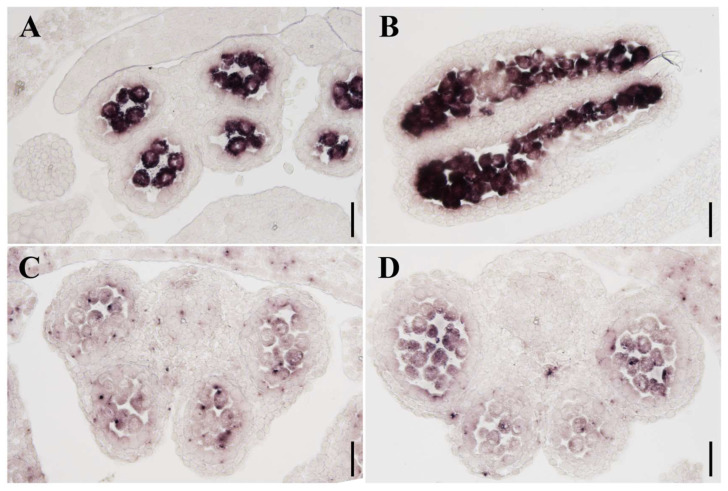
In situ hybridization analyses of the *PsPHD11* (**A**,**B**) and *PsPHD16* (**C**,**D**) genes in the wild-type anther of pea. The transverse sections of the flowers were made using the paraffin slice method, and the *PsPHD11* and *PsPHD16* transcriptional signals were detected in the developing anthers. (**A**–**D**) Scale bar = 50 um.

**Table 1 plants-13-01489-t001:** Basic information of PHD finger families in other species.

Species	Length (aa)	Molecular Weight (KDa)	pI	Family Members	Subfamily	Chromosome	References
*Pyrus bretschneideri*	-	-	-	31	10	17	[12]
*Arabidopsis thaliana*	234–2242	-	-	70	17	5	[8]
*Brassica rapa*	224–1748	25.45–192.55	4.70–8.99	33	5	10	[7]
*Medicago truncatula*	196–2371	22.87–258.81	4.81–9.33	64	14	8	[9]
*Glycine max*	132–2335	14.76–259.27	4.76–9.96	95	20	20	[10]
*Sorghum bicolor*	79–2265	8.67–249.88	4.22–9.23	79	6	10	[13]
*Triticum aestivum*	216–2853	24.57–310.35	4.42–9.65	244	4	42	[16]
*Oryza sativa*	175–2275	-	-	44	10	12	[15]
*Zea mays*.	72–2379	-	-	67	10	10	[14]
*Capsicum annuum*	216–2336	28.4–260.6	-	73	12	19	[11]

## Data Availability

Data is contained within the article or Appendix A.

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
