# Peer review of "Genome-Wide Identification and Expression Analysis of the PHD Finger Gene Family in Pea (Pisum sativum)"

_plants, 2024, doi:10.3390/plants13111489_

Round 1
Reviewer 1 Report
Comments and Suggestions for Authors
Peas are an important crop, and conducting all tasks associated with it holds significant value. This study conducted a genome-wide identification and expression analysis of the PHD finger family in pea, aiming to provide a theoretical reference for the biological function study of the PHD finger protein in pea.
Remark: Table 1 present as a Supplementary file
Reviewer 2 Report
Comments and Suggestions for Authors
I consider that these results could be a reference to study the functions of PHD finger family genes in pea. Minor changes regarding italics in case of scientific names or the use of capitalization in tables could be improved.
Reviewer 3 Report
Comments and Suggestions for Authors
This manuscript mainly provide the genome-wide identification of PHD family members in pea. All the members were clustered into 20 subfamilies and their gene structure, conserved domains, subcellular localization, and some physical and chemical properties were further analyzed. They also performed collinearity analysis in several species. The spatiotemporal expression pattern of the PsPHD members of the subfamily M during anther development were comprehansively analyzed by Qpcr and in situ hybridization.
Major comments:
1. The figure legend of all Figures must be clarified. All acronyms indicated in the figured must be clarified in the legend.
2. Line 303 Please provide clarification on the selection of subfamily M for further analysis.
3. Line 323 Please provide a reasonable explanation for why they focus on the anther development. The reasons provided currently are not clear.
4. In Fig. 4, please provide the information of the subfamilies in the phylogenetic tree and make it easier to understand the correlation between the subfamily category and their protein/gene structure. And please provide more detailed statements for them.
5. The synteny analysis of PsPHDs and their description was dryasdust and informationless.
Minor comments:
Line49-50, “The PHD finger” -Abbreviations should be provided in their full forms upon their first appearance.
Line 53-55, “PHDs” may not make sense.
Line 57-59, “PHDs genes” should be corrected.
Line 180-181 and the same cases in the title of Fig. 3, “genes” should be “proteins”.
Line 174 and similar cases throughout the paper: please change “sub-families” to “subfamilies”.
Line 324-325, “literature reports”, Please rephrase it.
Line 477-479 Which control gene was used in qPCR analysis? Please add that in Methods.
Line 306-307, 336-337, 383-384, "Suggesting that ..." the syntax errors are awkward.
Comments on the Quality of English Language
Extensive editing of English language required
